# Ablation of GPR56 Causes β-Cell Dysfunction by ATP Loss through Mistargeting of Mitochondrial VDAC1 to the Plasma Membrane

**DOI:** 10.3390/biom13030557

**Published:** 2023-03-18

**Authors:** Israa Mohammad Al-Amily, Marie Sjögren, Pontus Duner, Mohammad Tariq, Claes B. Wollheim, Albert Salehi

**Affiliations:** 1Department of Clinical Science, SUS, Division of Islet Cell Physiology, University of Lund, SE-205 02 Malmö, Sweden; 2Department of Clinical Science, SUS, Division of Experimental Cardiovascular Research, Lund University, SE-221 00 Lund, Sweden

**Keywords:** pancreatic islets, GPCRs, cAMP, ATP release, VDAC1, VDAC2, confocal microscopy and cellular signalling cytokine release, stress kinases

## Abstract

The activation of G Protein-Coupled Receptor 56 (GPR56), also referred to as Adhesion G-Protein-Coupled Ceceptor G1 (ADGRG1), by Collagen Type III (Coll III) prompts cell growth, proliferation, and survival, among other attributes. We investigated the signaling cascades mediating this functional effect in relation to the mitochondrial outer membrane voltage-dependent anion Channel-1 (VDAC1) expression in pancreatic β-cells. GPR56KD attenuated the Coll III-induced suppression of P70S6K, JNK, AKT, NFκB, STAT3, and STAT5 phosphorylation/activity in INS-1 cells cultured at 20 mM glucose (glucotoxicity) for 72 h. GPR56-KD also increased Chrebp, Txnip, and Vdac1 while decreasing Vdac2 mRNA expression. In GPR56-KD islet β-cells, Vdac1 was co-localized with SNAP-25, demonstrating its plasma membrane translocation. This resulted in ATP loss, reduced cAMP production and impaired glucose-stimulated insulin secretion (GSIS) in INS-1 and human EndoC βH1 cells. The latter defects were reversed by an acute inhibition of VDAC1 with an antibody or the VDAC1 inhibitor VBIT-4. We demonstrate that Coll III potentiates GSIS by increasing cAMP and preserving β-cell functionality under glucotoxic conditions in a GPR56-dependent manner by attenuating the inflammatory response. These results emphasize GPR56 and VDAC1 as drug targets in conditions with impaired β-cell function.

## 1. Introduction

The prevalence of metabolic diseases, in particular obesity and Type 2 diabetes (T2D), is high among the older population in western countries and is also increasing among the young around the globe. A resistance to insulin in its target organs, together with pancreatic β-cell dysfunction, plays a central role in the development of the metabolic syndrome and T2D [1,2,3]. Many studies have shown that genetic and environmental factors are interconnected in promoting the development of the disease through the failure of the β-cells to increase insulin secretion in compensation for resistance to the hormone [1,3]. In most of these conditions, suboptimal blood glucose control and dyslipidemia during years of prediabetes lead to β-cell dysfunction, dedifferentiation, and, eventually, apoptosis [1,4,5]. Thus, the prevention of β-cell dysfunction would have a great impact on the prevention of the metabolic syndrome and T2D development [5,6,7].

Glucotoxicity induces cellular stress in β-cells, such as the activation of inflammatory signals, oxidative stress, and endoplasmic reticulum stress, ultimately impairing the functions of vital subcellular organelles, including mitochondria [8,9]. Since β-cell mitochondria play a central role in the coupling of glucose metabolism to insulin secretion, their dysfunction has been implicated in the defective hormone release in T2D [8,9,10]. Mitochondrial metabolism requires an inward and outward flux of hydrophilic metabolites, including ATP, ADP, and respiratory substrates, through voltage-dependent anion channels (VDACs), also known as mitochondrial porins, in the mitochondrial outer membrane [11]. There are three VDAC isoforms (VDAC1, VDAC2, and VDAC3) differentially expressed in mammalian tissues [11,12]. We previously presented evidence that pancreatic β-cells express all three isoforms [13,14]. We described the overexpression of VDAC1 and downregulation of VDAC2 in islets of T2D organ donors. This causes a mis-targeting of VDAC1 to the β-cell surface, resulting in ATP loss and impaired glucose-stimulated insulin secretion (GSIS). This paradigm is mimicked by the culture of human islets in the presence of high glucose and can be prevented by VDAC1 knock-down, or by an acute addition of VDAC1 inhibitors [14]. These results show that T2D β-cells are dysfunctional, and that GSIS can be recovered both in vitro [14] and in vivo [7,15].

G-Protein-Coupled Receptors (GPCRs) constitute the largest group of cell surface receptors in men and are also the targets of ~35% of all prescription medicines [16,17]. G-Protein-Coupled Receptor 56 (GPR56), also known as Adhesion G-Protein-Coupled Receptor G1 (ADGRG1), is highly expressed in many tissues, including pancreatic β-cells [18,19,20,21]. Functionally, GPR56 has been linked to a myriad of biological and physiological processes within the cells, including the development and differentiation of cerebral cortex, innate immunity, muscle cell and oligodendrocyte development, and carcinogenesis, as well as affecting pancreatic β-cell function [18,19,20,21,22,23]. The GPR56 receptor is an adhesion receptor characterized by an extremely long N-terminal extracellular domain (NTD), which is cleaved off from the seven-membrane-spanning C-terminal domain (CTD) by auto-proteolysis upon binding to its principal ligand Collagen Type III. The CTD initiates signaling, the nature of which depends on the cellular context. Several studies have shown GPR56 coupling to the Gα12/13 class of heterotrimeric G-proteins to promote RhoA activation and alterations of the actin filamentous network [22,23]. GPR56 has also been reported to signal via Gαs to stimulate adenylate cyclase [18,24], as well as raising [Ca^2+^]_i_ through Gαq [19], although this was not clearly linked to Gαq since it is in contrast to ATP-induced [Ca^2+^]_i_ rise, which was abolished in the absence of extracellular Ca^2+^ [19], suggesting the activation of calcium channels [20,21]. We have previously shown that GPR56 activation promotes β-cell viability and function both in human and rodents; in human pancreatic β-cells, GPR56 expression correlates with the expression levels of transcripts, vital for the function of β-cells and GPR56 mRNA, which is indeed reduced in islets of Type 2 diabetic organ donors [18]. We have also shown that the knock-down of GPR56 (GPR56-KD) leads to β-cell dysfunction, reduced cell viability, and attenuates the beneficial effect of Collagen Type III (Coll III) on β-cell function [18]. Our observations thus corroborate reported findings showing that, although GPR56-KO mice have normal islet vascularization and only mildly impaired glucose tolerance, the activation of GPR56 by Coll III increases islet insulin secretion and enhances cell viability [19,25]. GPR56 has, thus, been mainly linked to protecting β-cells from apoptosis, but it is similarly important for the insulin secretory function of β-cells. GPR56 is linked to the activation of the cAMP/Protein Kinase A (PKA) system [18], while it is well-known that the cAMP- and calcium-signaling pathways are coupled, i.e., a rise in [Ca^2+^]_i_ can also activate adenylate cyclase [15,26]. These results show the complexity of GPR56-induced signaling pathways, which are not completely elucidated. By virtue of its transmembrane expression and its capability of also having adhesive GPCR functions, GPR56 seems not only to regulate secretory capacity but also interlinking many intracellular-signaling pathways important for β-cell structure and viability.

The aim of the present investigation was to study to what extent GPR56-KD could interfere with the effect of Coll III on the activity of several important stress kinases, and whether GPR56-KD is associated with increased VDAC1 expression and its mistargeting to the cell surface in rodent and human β-cells.

## 2. Materials and Methods

### 2.1. Animals

Female mice (C57/bl) (Janvier Laboratory, Saint Isle, France) weighing 25–30 g were used under standard conditions (12 h light/dark cycle, 22 °C) with access to standard pellet diet (B&K) and water ad libitum. The study protocol was approved by the Ethics Committee for Animal Research at Lund University (1057/2020). Isolation of pancreatic islets was performed by retrograde injection of a collagenase solution via the pancreatic duct and islets, which were then collected under a stereomicroscope at room temperature [27].

### 2.2. Reagents

Fatty acid-free bovine serum albumin (Boehringer, Ingelheim, Germany), rabbit polyclonal and mouse monoclonal anti-VDAC1 antibody (N-terminal) (Abcam, Cambridge, UK and Santa Cruz Biotechnology Inc, Santa Cruz, CA, USA respectively), insulin ELISA kit (Mercordia, Uppsala, Sweden), primers and qPCR materials were from Applied Biosystems (Waltham, MA, USA). Cell Signaling Multiplex Assay from Merck Millipore, VBIT-4 and AKOS022075291 (AKOS) were from Glixx Laboratories (Hopkinton, MA, USA). All other chemicals were from Merck AG (Darmstadt, Germany) or Sigma (Saint Louis, MO, USA).

### 2.3. INS-1 832/13 Cell Culture

INS-1 832/13 cells (kindly donated by Dr. C. B. Newgaard, Duke University, Durham, NC, USA) were cultured in RPMI-1640 containing 11.1 mM of D-glucose and supplemented with 10% fetal bovine serum, 100 U/mL of penicillin (Gibco, BRL, Gaithersburg, MD, USA), 100 μg/mL of streptomycin (PAA Laboratories, Toronto, Ontario, Canada), 10 mM of N-2 hydroxyethylpiperazine-N’-2-ethanesulfonic acid (HEPES), 2 mM of glutamine, 1 mM of sodium pyruvate, and 50 μM of β-mercaptoethanol (Sigma Aldrich, Saint Louis, MO, USA) at 37 °C in a humidified atmosphere containing 95% air and 5% CO2. For an assessment of the effects of elevated glucose (glucotoxicity), cells were treated with 20 mM of D-glucose (20G) for 72 h. Controls were maintained in 5 mM of glucose (5G) media. For GPR56 knockdown by siRNA, INS1 832/13 cells were cultured to 75% confluency and then subjected to GPR56-KD as previously described [18]. The siRNAs used for Gpr56-KD are shown in Appendix A. Each treatment was carried out in three biological replicates, along with scramble controls. Thereafter, the cells were washed and cultured for 6 h in a normal RPMI medium (recovery period) before being cultured in RPMI-1640 with 5 or 20 mM of glucose (5 G or 20 G) in the presence or absence of 20 ug/mL of Collagen-III (Col III) for 72 h (Figure 1), or they were subjected to incubation with indicated agents (see results Section 3.4).

### 2.4. EndoC βH1 Cell Culture

The human clonal β-cell line EndoC-βH1 (EndoCells, Paris, France) was seeded in 24 well plates at a density of 1.8 × 105 cells/well and maintained in DMEM culture medium (5.5 mM glucose) 2% BSA fraction V (Roche, Basel, Switzerland), 10 mM of nicotinamide (Merck, Darmstadt, Germany), 50 µM of 2-mercaptoethanol, 5.5 µg/mL of transferrin, 6.7 ng/mL of sodium selenite (Sigma), 100 U/mL of penicillin, and 100 µg/mL of streptomycin (PAA Laboratories, Toronto, Ontario, Canada) as described in [28]. For measurement of insulin secretion, the cells were cultured for 12h in the culture medium containing 2.8 mM glucose before incubation at 1 or 20 mM of glucose.

### 2.5. Knockdown of GPR56 in Mouse Islets, INS-1 832/13, and EndoC βH1 Cells

GPR56-KD in mouse islets was performed using a cocktail of three different Lentivirus delivered by shRNAs targeting the GPR56 gene (Santa Cruz, CA, USA), as described previously in [18]. For downregulation of GPR56 in INS-1 832/13 cells and EndoC βH1 cells, siRNA from ThermoFisher Scientific (Wilmington, DE, USA), with an appropriate scrambled control, were used according to the manufacturer’s recommendations. Validation of GPR56-KD was determined by qPCR and confocal microscopy, as also described previously [18]. The siRNAs, or shRNAs used for GPR56-KD, are shown in Appendix A.

### 2.6. Determination of Intracellular Pathways

The Effect of Collagen Type III (Col III) on the major signaling pathways was determined in scramble control (Scr control) and Gpr56-KD INS-1 832/13 cells cultured at 5 or 20 mM of glucose for 72 h. Detection of phosphorylated P70S6K, JNK, AKT, NFκb, STAT3, and STAT5 was assayed on the cell extracts by Luminex, according to the manufacturer’s protocol.

### 2.7. Immunostaining and Confocal Imaging

Isolated mouse islets, as well as EndoC-βH1 cells, were seeded on glass-bottom dishes cultured overnight. They were then washed twice and fixed with 3% paraformaldehyde for 10 min, followed by permeabilization with 0.1% Triton-X 100 for 15 min. The blocking solution contained 5% normal donkey serum in PBS and was used for 15 min. Primary antibodies against VDAC1 (Abcam, 1:200), SNAP-25 (Abcam, 1:100), Na^+^/K^+^ ATPase (Abcam, 1:100), and Guinea pig insulin (Eurodiagnostica, 1:300) were diluted in blocking buffer and incubated overnight at 4 °C. Immunoreactivity was quantified using fluorescently labeled secondary antibodies (1:200) and visualized by confocal microscopy (Carl Zeiss, Germany).

### 2.8. Quantitative Polymerase Chain Reaction (qPCR)

Total RNA was extracted from INS1 832/13 and EndoC-βH1 cells using RNAeasy (Qiagen, Hilden, Germany) before complementary DNA (cDNA) was synthesized using SuperScript (Invitrogen, Carlsbad, CA, USA), according to the manufacturer’s protocol. Concentration and purity of total RNA were measured with a NanoDrop ND-1000 spectrophotometer (A260/A280 > 1.9 and A260/A23 0 > 1.4) (NanoDrop Technologies LLC, Wilmington, DE, USA). RNA Quality Indicator (RQI) higher than 8.0 (Experion Automated Electrophoresis, Bio-Rad, USA) was considered to be high-quality total RNA preparation. TaqMan mastermix from Applied Biosystems (Foster City, CA, USA) was used for qPCR and performed following manufacturer’s protocol and was run in a 7900 HT Fast Real-Time System (Applied Biosystems). The qPCR was carried out as follows: 50 °C for 2 min, 95 °C for 10 min, 40 cycles of 95 °C for 15 s, and 60 °C for 1 min. Changes in gene expression were calculated using the ΔΔCt method with a fold-change cut-off at ≥ 1.5 and *p* < 0.05 considered significant. All samples were run in duplicate, and relevant negative controls were run on each plate. qPCR results were normalized to housekeeping genes (PPIA or HPRT). Primer sequences used in the qPCR assays are provided in Appendix A.

### 2.9. Western Blots

EndoC βH1 cells were homogenized in ice-cold RIPA buffer and kept shaking on ice for 30 min. Extracted total protein content from homogenates was measured by Pierce BCA Protein Assay Kit (Thermo Scientific, Waltham, MA USA). Homogenate samples (10 µg) from scramble control (Scr) or GPR56-KD cells were electrophoresed on 7.5% SDS-polyacrylamide gel (Bio-Rad, Hercules, CA, USA). After electrophoresis, proteins were transferred to a nitrocellulose membrane (Bio-Rad). The membrane was blocked in LS-buffer (10 mmol/L Tris, pH 7.4, 100 mmol/L NaCl, 0.1% Tween-20) containing 5% non-fat dry milk powder for 60 min at room temperature. Subsequently, the membranes were incubated overnight with the same VDAC1 antibody used for the confocal experiments (1:10,000) at 4 °C. After washing (three times) in LS-buffer, the membrane was finally incubated with a horseradish peroxidase-conjugated anti-rabbit antibody (1:5000) (Bio-Rad, Hercules, CA, USA). Detection of α-tubulin was with rabbit-anti-α-tubulin (Sigma, USA) and secondary anti-rabbit anti-body (1:1000). Immunoreactivity was detected using an enhanced chemiluminescence reaction (Pierce, Rockford, IL, USA). The blots were scanned with ChemiDocTM MP Imaging System (Bio-Rad), and bands corresponding to the ~37-kDa (protein marker) were identified as VDAC1 protein. A typical Western blot image of entire gel, performed on the homogenates from two Scr controls and four GPR56-KD cells, is shown in Appendix A.

### 2.10. Insulin Secretion

For functional studies after recovery, the siRNA-treated INS-1 832/13, or EndoC βH1 cells, were washed and pre-incubated for 120 min at 37 °C in SAB buffer, pH 7.4, and supplemented with 10 mM of HEPES, 0.1 % bovine serum albumin, and 2.8 mM of glucose. After pre-incubation, the buffer was changed and INS-1 832/13, or EndoC βH1 cells, were incubated at 1 or 20 mM of glucose with indicated agents for 60 min at 37 °C. Immediately after incubation, an aliquot of the medium was removed and frozen for subsequent assay of insulin. The cells were then washed with PBS and stored in 100 mM of HCl containing IBMX (100 μM) for subsequent analysis of cyclic AMP.

### 2.11. cAMP Determination

The cAMP content in the cell lysate was measured using a direct cAMP ELISA kit (Enzo Life Sciences, Farmingdale, NY, USA) according to the manufacturer’s instructions, and the values were related to protein content. The protein concentrations of the cell lysates were measured by a BCA kit (Thermo Fisher Scientific, Wilmington, DE, USA).

### 2.12. ATP Determination

ATP content (INS-1 832/13 and EndoC βH1 cells) and release (INS-1 832/13 cells) in incubated cells after GPR56-KD were determined using a luminometric assay kit according to manufacturer’s recommendation (BioVision, Milpitas, CA, USA) and normalized to protein content. After incubation, the cells were washed with PBS buffer (three times) in Ripa buffer containing protease inhibitors and stored at −80 °C for subsequent measurements of cellular ATP, while the released ATP was measured in the 1-h incubation medium. The protein contents of each sample were analyzed by BCA protein kit (Thermo Scientific, IL, USA).

### 2.13. Statistics

The results are expressed as means ± SEM for the indicated number of observations or illustrated by an observation representative of the results obtained from different experiments (confocal microscopy). The significance of random differences was analyzed by Student’s *t*-test or, where applicable, an analysis of variance was performed, followed by Tukey–Kramers’ multiple comparisons test. *p*-value < 0.05 was considered significant.

## 3. Results

### 3.1. The Consequence of GPR56-KD on the Activation of Several Intracellular Pathways

The impact of long-term high glucose (20 mM, 72 h) culture on the key signaling molecules (pathways) involved in the physiology/pathophysiology of β-cells in scramble control INS-1 cells in the presence or absence of the naturally occurring GPR56 agonist Coll III was studied. The effect of Coll III on these signaling molecules was only investigated in Gpr56-KD cells under the two glucose culture conditions. Cell-signaling analysis revealed an increase in p-P70S6K, p-JNK, and p-NFκB induced by high glucose, while p-AKT, p-STAT3, and p-STAT5 were not significantly altered in Scr control INS-1 cells (Figure 1A–C). The presence of Coll III (20 µg/mL) during the culture period suppressed the phosphorylation of P70S6K, JNK, NFκB, AKT, STAT3, and STAT5, both at 5 or 20 mM of glucose in Scr control cells (Figure 1A–F). This effect of Coll III was abolished in Gpr56-KD INS-1 cells regardless of ambient glucose levels (Figure 1A–F). While high glucose did not appreciably affect p-CREB, it was increased in scramble control INS-1 832/13 cells by the presence of Coll III (not shown). Coll III actions are thus mediated through GPR56 activation.

### 3.2. The Effect of GPR56-KD on the Expression of Chrebp, Txnip, Vdac1 and Vdac2

Since we have reported that GPR56-KD is associated with β-cell dysfunction and decreased viability reminiscent of diabetic β-cells [18], we evaluated the impact of GPR56-KD on mitochondrial VDAC1 and VDAC2 expression, as we have demonstrated that the altered expression of VDAC1 and VDAC2 is associated with mitochondrial dysfunction, leading to impaired β-cell function [13,14]. We also assessed the consequence of GPR56-KD on Chrebp and txnip, two transcriptional factors of importance for the glucotoxicity-induced increase in VDAC1 expression in pancreatic β-cells [14]. As seen in Figure 2, GPR56-KD significantly increased mRNA expression of Chrebp (A), Txnip (B), and Vdac1 (C), while Vdac2 mRNA was reduced (D). The calculated efficiency of GPR56-KD showed a reduction of almost 75% of GPR56 mRNA compared to the scramble control group (Appendix A). To evaluate whether GPR56-KD would have any off-target effect, we have also analyzed the mRNA level of a highly expressed GPCR in β-cells i.e., Gprc5b [29,30]. The mRNA level of Gprc5b was not affected by Gpr56-KD, revealing no off-target effect (Appendix A).

### 3.3. The Impact of Gpr56-KD on the Vdac1 Protein Expression and its Mistargeting to the Cell Surface in Mouse Islets

Since increased Vdac1 is associated with its mistargeting to the β-cell membrane contributing to β-cell decompensation [14], we next investigated the impact of Gpr56-KD on protein expression and sub-cellular localization of Vdac1 in isolated mouse pancreatic islets by confocal microscopy. Compared with the very low Vdac1 expression in scramble control islets (Figure 3A), Gpr56-KD markedly increased Vdac1 protein expression in the islets (Figure 3B). We studied the co-localization of Vdac1 with the plasma membrane-associated SNARE protein SNAP-25 (cell membrane marker) in islets after Gpr56-KD. Remarkably, confocal microscopy revealed Vdac1 co-localization with SNAP-25 in insulin positive cells, indicating Vdac1 surface localization in β-cells of Gpr56-KD islets (Figure 3B).

### 3.4. The Impact of Gpr56-KD on cAMP, ATP Content, and Insulin Secretion

Vdac1 is an ATP-conducting anion channel normally allowing the transport of ATP from mitochondria to the cytoplasm in cells [11,14]. Since immunohistochemical experiments showed an increased VDAC1 expression in pancreatic β-cells upon Gpr56-KD, we next evaluated the effect of two different VDAC1 blockers on the cellular cAMP and ATP content in relation to GSIS in scramble control and Gpr56-KD INS-1 832/13 cells. In the following experiments, the cAMP and ATP content, as well as the GSIS, were measured after 60 min incubation of INS-1 cells in the presence or absence of Coll III and VDAC1 inhibitors, i.e., VBIT-4 and VDAC1 antibodies (VD1ab) [14,31]. As shown in Figure 4A, the increased cellular cAMP content induced by high glucose was further augmented by Coll III, but not by the presence of VBIT-4 or VD1-ab during the short-term incubation of scramble control cells. In Gpr56-KD cells, both basal- and glucose-stimulated increases in cAMP were diminished, and the presence of Coll III did not alter cAMP generation. In contrast, the inhibition of VDAC1 with VBIT-4 and VDAC1 antibodies further increased glucose-stimulated cAMP generation in the KD cells. Our data also show that a glucose-stimulated increase in cellular ATP level was not affected by Coll III or by VBIT-4 and VDAC1 antibodies in scramble control INS-1 832/13 cells. In Gpr56-KD cells, the diminished glucose-induced rise in ATP level was restored only in the presence of VBIT-4 and VDAC1 antibodies (Figure 4B).

To evaluate the impact of altered cellular signaling, we monitored GSIS. Insulin secretion was potentiated by Coll III, while, as expected, VBIT-4 or VDAC1-ab did not alter secretion in scramble control INS-1 832/13 cells. In Gpr56-KD cells, the GSIS was markedly reduced and Coll III potentiation was abolished. The presence of VBIT-4 and VDAC1-ab during the final incubation improved GSIS (Figure 4C).

As mistargeting of VDAC1 to the plasma membrane causes loss of ATP [14], we next investigated the impact of GPR56-KD on the ATP release in INS-1 cells at 1 mM glucose to avoid high glucose-induced ATP generation [14]. After GPR56-KD, the INS-1 cells were incubated for 60 min at 1 mM of glucose in the presence or absence of Coll III, VBIT-4, and AKOS, which is another VDAC1 inhibitor [14,31]. The basal ATP release from scramble control INS-1 cells was not affected by Coll III, VBIT-4, or AKOS (Figure 5). However, Gpr56-KD was associated with a markedly increased ATP loss from INS-1 832/13 cells (Figure 5), which was not affected by the presence of Coll III. It is noteworthy that the ATP release was markedly prevented by VBIT-4 and AKOS (Figure 5).

Finally, we extended the observations in INS-1 cells to human EndoC βH1 cells, studying the impact of GPR56-KD on VDAC1 expression, apoptosis, and the release of inflammatory cytokines, as well as cAMP generation and insulin secretion. Among a panel of cytokines (IL-2, IL-6, IL-10, IFNγ, IL-12bp40, IL-12p70, and IL-17) that were analyzed, as seen in Figure 6A–C, GPR56-KD was associated with an increased release of MCP-1 (CCL2), IL-2, and TNF-α from EndoC βH1 cells, while the release of other measured cytokines was undetectable (not shown) in scramble control or GPR56-KD cells. Western blot and immunohistochemical analysis by confocal microscopy revealed that GPR56-KD was associated with increased VDAC1 protein expression (Appendix A). GPR56-KD clearly caused VDAC1 mistargeting to the cell surface, as revealed by co-staining with Na^+^/K^+^ATPase in EndoC βH1 cells (Figure 6D,E). GPR56-KD also resulted in an increased intensity of nuclear Hoechst staining, indicating an increased apoptotic rate [32] (Figure 6F). Interestingly, GPR56-KD (Appendix A) in EndoC βH1 cells attenuated both GSIS and glucose-induced increase in cAMP when the cells were incubated for 60 min at 20 mM glucose, while basal insulin release or cAMP content was not influenced (Figure 6G,H). The potentiation of GSIS concomitant with the cAMP generation by Coll III was also markedly attenuated by GPR56-KD (Figure 6G,H). These results link β-cell stress after GPR56 loss-of-function to VDAC1-mediated abrogation of ATP formation and the subsequent impairment of stimulus-secretion coupling.

## 4. Discussion

The rationale for the present investigations is the documented role of GPR56 in pancreatic β-cell survival and secretory function [18,19,21,25,33]. The expression level of GPR56 is positively correlated with the transcript level of a great number of genes with a beneficial impact on the β-cell fate in human pancreatic islets [18]. Moreover, GPR56 is downregulated in islets from T2D organ donors [18]. Herein, we define the mechanism by which GPR56 loss-of-function causes pancreatic β-cell dysfunctionality.

As GPR56 is the most abundant GPR both in mouse and human β-cells [29], it is of interest that not only Coll III derived from islet endothelial cells [19], but also the insulinotropic amino acid L-phenylalanine has been identified as a GPR56 agonist [34]. The stimulation of insulin secretion by L-phenylalanine, a ligand for GPR142, is preserved in GPR142 KO islets, which could be explained by GPR56-mediated signaling [19,35]. The cumulating evidence makes GPR56 an interesting drug target in T2D. We and others have presented evidence that GPR56 is indeed capable of acting as a G-protein-coupling (Gαs type) to elicit downstream signaling cascades via the cAMP/PKA system in pancreatic β-cells [18,21,33], which was confirmed in the present work. A similar signal transduction has been shown for GpR56 activation by testosterone in prostate cells [24]. Although signaling via Gαq has also been reported in β-cells and neuronal cells [19,20,21,36], the cAMP/PKA system could act by lowering the threshold level of the exocytosis for intracellular Ca^2+^ ([Ca^2+^]_i_) elevation, increasing the secretory response of β-cell to even small increases in [Ca^2+^]_i_ [15].

In the current study, we present mechanistic data revealing GPR56 as a mediator of the suppressive effect of Coll III on cellular stress-related signals, such as P70S6K, JNK, AKT, NFκb, STAT3, and STAT5 signaling, thereby providing the pathway(s) by which GPR56 activation prevents β-cell dysfunction. While it is well-established that stress kinases, NFκB, STAT3, and STAT5 signaling is required for an array of physiological/pathophysiological events, they are mostly involved in stress- or inflammation-induced β-cell dysfunction [6,37]. We show here that GPR56-KD causes cellular stress that also results in the release of certain inflammatory cytokines, such as MCP-1, IL-2, and TNFα by the β-cells. This observation confirms previous studies that GPR56 plays a role in inflammation and has been identified as an inhibitory receptor, suppressing the pro-inflammatory activity of cytotoxic lymphocytes [38]. Likewise, cytokine-induced β-cell apoptosis is prevented by Coll III, but not by a simple overexpression of GPR56 [25]. However, the delineation of the inflammatory mechanism in GPR56-deficient β-cells extends beyond the scope of the present work and merits further investigation. It should also be mentioned that the altered regulation of NFκB, STAT3, and STAT5 plays a critical role in inducing/maintaining the chronic, low-grade inflammation that conveys both β-cell dysfunction [37] and complications of diabetes, including atherosclerotic vascular lesions [39] by influencing diverse cellular gene expression programs.

Moreover, we link GPR56-KD to increased expression of the transcription factors Chrebp and Txnip that initiate the overexpression of VDAC1 with a consequent reduction of VDAC2 expression in INS-1 832/13 cells. Chrebp and Txnip are highly increased in pancreatic β-cells in glucotoxic condition [40] and in islets from T2D organ donors [14,41]. TXNIP is known to activate the NLRP3 inflammasome, generating interleukin-1β, thereby contributing to impaired β-cell function [37,42]. An increased cAMP formation exerts β-cell protection in part by promoting TXNIP proteosomal degradation [42], an effect that could explain coll III protection from cytokine-induced apoptosis in human islets [25]. The overexpression of VDAC1 and its mistargeting to the β-cell plasma membrane leads to a loss of ATP, the crucial metabolic coupling factor in GSIS [8]. We have shown previously that the prevention of ATP loss by the acute addition of VDAC1 antibodies and inhibitors completely restored the defective GSIS in islets from human T2D organ donors and diabetic db/db mice [14].

Remarkably, confocal microscopy revealed that VDAC1 surface mistargeting occurs in GPR56-KD mouse islet β-cells, as shown by the co-localization with the plasma membrane-associated SNARE protein SNAP-25. A similar VDAC1 mistargeting was observed after GPR56 KD in human EndoC βH1 cells. It is noteworthy that VBIT-4 and AKOS, two chemical VDAC1 inhibitors [14,31], as well as the VDAC1 antibody efficiently restored glucose-induced rises in cellular ATP levels, the generation of cAMP, and the stimulation of insulin secretion in the GPR56 KD cells. We show in INS-1 GPR56-KD cells that this is mainly due to the attenuation of the high rate of ATP leakage. The most plausible explanation for the acute restoration of cAMP generation is the increase in cellular ATP, which drives the cAMP formation during glucose stimulation [43]. In contrast, the stimulatory effects of Coll III are not restored, further validating GPR56 as the collagen receptor. Plasma membrane-resident gated VDAC1 has been documented in various mouse and human tissues with the mitochondrial surface residues facing the extracellular space [44,45,46]. Of note, oxidative stress in neurons activates the conductance of the neurolemmal VDAC1, initiating apoptosis, which is prevented by antibodies directed against the extracellular N-terminus of VDAC1 [45]. Interestingly, the association of GPR56 deletion with the mistargeting of mitochondrial VDAC1 to the cell surface further emphasizes the importance of GPR56 in the regulation of insulin secretion. Loss of GRP56 function in T2D [18] thus participates in altered gene expression and β-cell dysfunction. At first sight, this conclusion seems to be at variance with the only mild glucose intolerance of the GPR56 KO mouse [19]. We speculate that the deletion of GPR56 during fetal development may upregulate other adhesion GPRs, which would not necessarily occur during short-term KD of the receptor.

VDAC1 upregulation and oligomerization is caused by oxidative and nitrosative stress, not only in β-cells in T2D but also in neurodegenerative diseases, in particular, Alzheimer’s disease [47,48]. It is, therefore, of great interest that VBIT-4, which inhibits VDAC1 conductance and oligomerization, prevents onset of diabetes in db/db mice [14] and markedly improves the phenotype in a mouse model of Alzheimer’s disease [48]. VBIT-4, thus, has both acute effects on cell signaling by preventing ATP loss through VDAC1 expressed in the plasma membrane and long-term actions on gene expression and cell function in diseases linked to oxidative stress and mitochondrial dysfunction.

Taken together, the present data show that GPR56 activation by Coll Type III is associated with the suppression of P70S6K, JNK, AKT, NFκb, STAT3, and STAT5 phosphorylation/activity and increased CREB signaling. Since GPR56 positively modulates the activity of the cAMP-PKA system in the β-cell ([18] and present work), the Coll Type III-mediated beneficial effects on the β-cell function seem to be through cAMP signaling concomitant with the suppression of the aforementioned stress kinases and transcriptional factors. A further signaling pathway implicating GPR56 is its link to integrin function [20,21]. This is relevant for β-cell senescence in which, among others, STAT3 is upregulated [49].

## 5. Conclusions

In the current work, we have linked the dysregulation and mistargeting of the diabetes executer protein VDAC1 [14] to the suppression of GPR56, thereby linking this adhesion GPCR to β-cell mitochondrial dysfunction with impaired ATP accumulation and compromised insulin secretion. GPR56 could, therefore, constitute a novel drug target preventing the loss of β-cell function in prediabetes and diabetes.

## Figures and Tables

**Figure 1 biomolecules-13-00557-f001:**
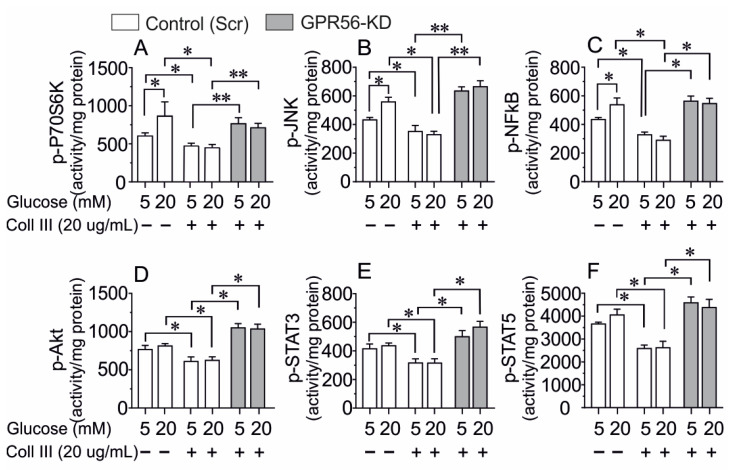
The Effect of Collagen type III (Col III) on the major signaling pathways in scramble control (Scr control) and Gpr56-KD INS-1 832/13 cells cultured at 5 or 20 mM glucose for 72 h. Col III suppressed the activity of P70S6K (**A**), JNK (**B**), AKT (**C**), NFκb (**D**), STAT3 (**E**), and STAT5 (**F**) in scr control but not in GPR56-KD cells. Detection of phosphorylated P70S6K, JNK, AKT, NFκb, STAT3, and STAT5 were assayed on the cell extracts by Luminx. The results are Mean±SEM for *n* = 6−8 different experiments in each group. * *p* < 0.05; ** *p* < 0.01.

**Figure 2 biomolecules-13-00557-f002:**
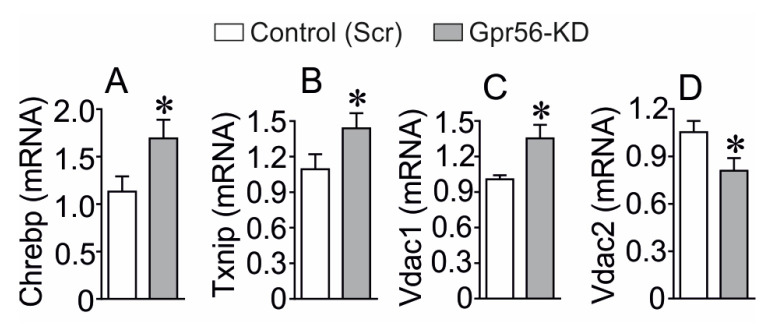
Effect of GPR56-KD on the expression of Chrebp, Txnip, Vdac1, and Vdac2 mRNA expression in INS-1 832/13 cells. GPR56-KD increased expression of Chrebp (**A**), Txnip (**B**), Vdac1 (**C**), and reduced Vdac2 (**D**), mRNA as measured by qPCR in INS-1 832/13 cells. Mean±SEM from four different experiments in each group are shown. * *p* < 0.05.

**Figure 3 biomolecules-13-00557-f003:**
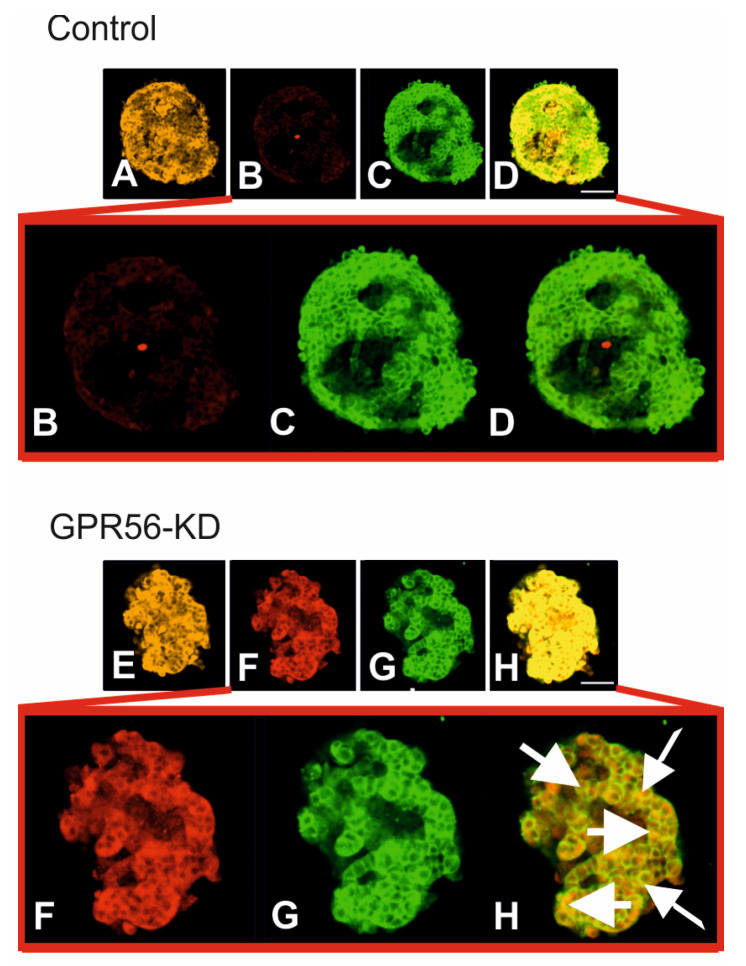
A representative confocal microscopy image of mouse islets double immune-labelled for insulin (beige) (**A**,**E**) and mitochondrial VDAC1 (red) (**B**,**F**), SNAP-25 (green) (**C**,**G**) and merged (**D**,**H**) in scramble control and Gpr56-KD mouse islets. Co-localization of VDAC1 and the membrane marker SNAP-25 is seen as orange-yellowish fluorescence in insulin-subtracted images. Bar indicates 20 µm. Increased magnifications of the same image (**B**–**D**) and (**F**–**H**) are also shown.

**Figure 4 biomolecules-13-00557-f004:**
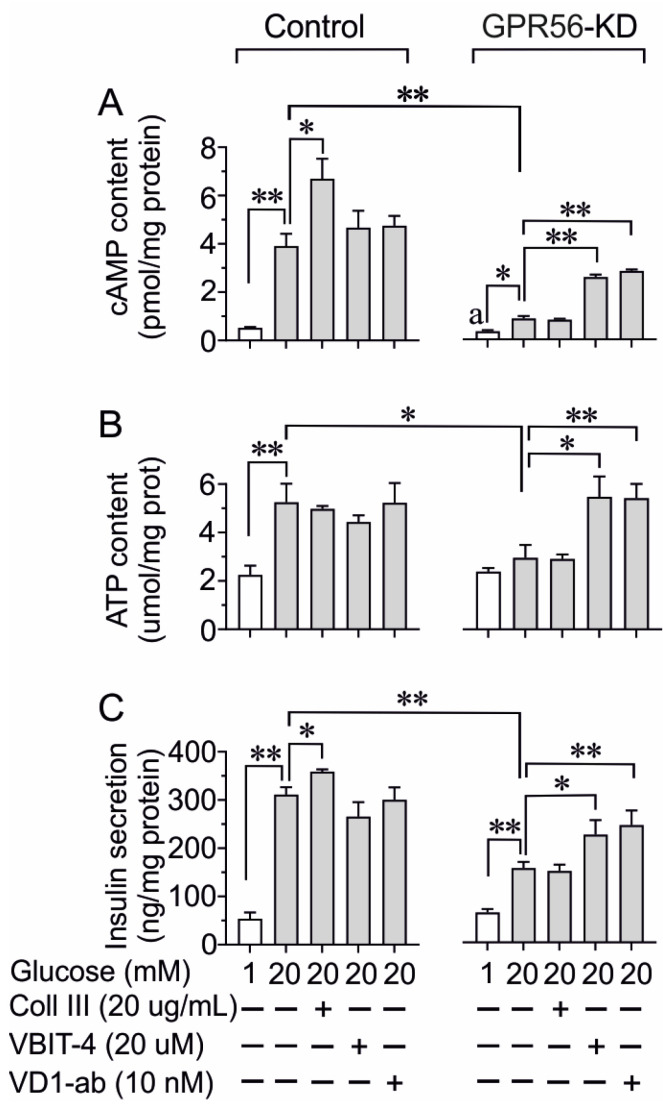
The effect of Coll III, VBIT-4, or VDAC1 antibodies (VD1 ab) on glucose (20 mM)-stimulated cAMP (**A**) and ATP content (**B**) as well as insulin secretion (**C**), in scramble control and GPR56-KD INS-1 832/13 cells. The cells were incubated at 1 or 20 mM glucose for 1 h. For comparisons cAMP and ATP content with insulin release, at basal glucose is included. Mean ± SEM for three experiments performed at different occasions (with three rep-licates in each group) are shown. *, *p* < 0.05; **, *p* < 0.01.

**Figure 5 biomolecules-13-00557-f005:**
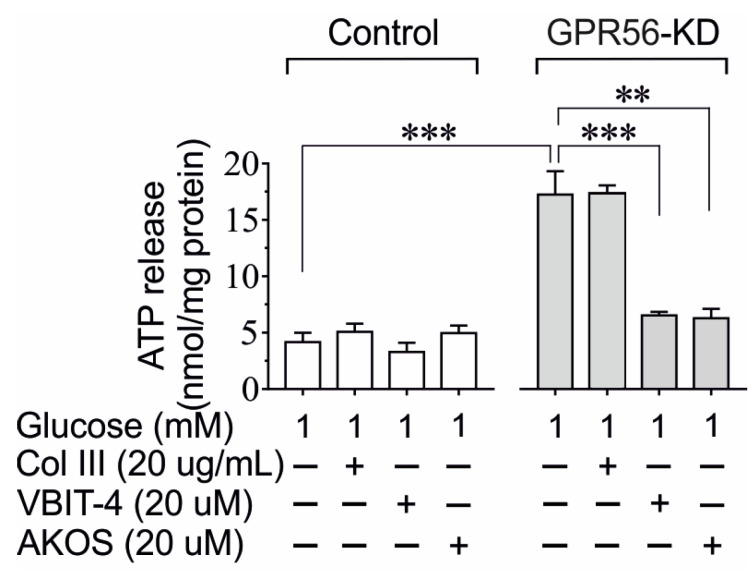
The effect of Coll III and VDAC1 inhibitors (VBIT-4 or AKOS) on ATP release in scramble control and Gpr56-KD INS-1 832/13 cells. The INS-1 832/13 cells were incubated at 1 mM of glucose in the presence or absence of Coll III, VBIT-4, or AKOS022075291 (AKOS) for 1 h. Mean ± SEM for *n* = 5 experiments performing at different occasions are shown. **, *p* < 0.01; ***, *p* < 0.001.

**Figure 6 biomolecules-13-00557-f006:**
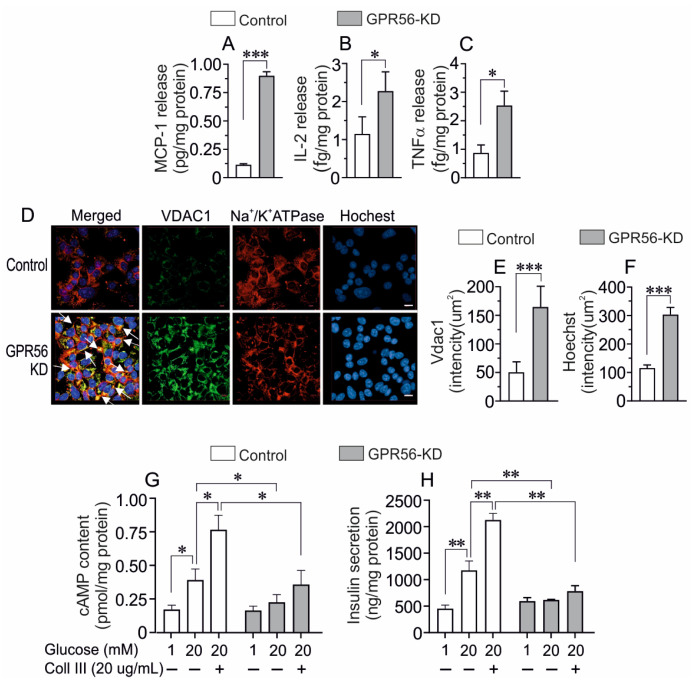
The effect of GPR56-KD on VDAC1 expression, cytokine release, cellular cAMP, and glu-cose-stimulated insulin secretion in human EndoC βH1 cells. GPR56-KD increased the release of MCP-1 (**A**), IL-2 (**B**), and TNFα (**C**) upon 6 h recovery culture from scramble control and GPR56-KD EndoC βH1 cells performed at six different occasions is shown. GPR56-KD increased VDAC1 expression and caused its mistargeting to the plasma membrane detected as co-localization with Na+/K+ ATPase, seen as orange-yellowish fluorescence (merged) concomitant with an increased Hoechst intensity in EndoC βH1 cells (**D**) VDAC1 (green), Na^+^/K^+^ ATPase (red), and Hoechst (blue). Calculation of VDAC1 intensity (**E**) and Hoechst intensity (reflecting apoptosis) (**F**) in EndoC βH1 cells shown in (**D**). The effect of Coll III on cAMP content (**G**) and glucose-stimulated insulin secretion (**H**) in scramble control and GPR56-KD EndoC βH1 cells incubated for 1h. The cAMP content and insulin release at basal glucose (1 mM) is included. Mean ± SEM for three experiments (*n* = 3) performed at different occasions with triplicates in each group are shown. * *p* < 0.05; ** *p* < 0.01, *** *p* < 0.005.

## Data Availability

All presented data is in the article.

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
