# Peer review of "Ablation of GPR56 Causes β-Cell Dysfunction by ATP Loss through Mistargeting of Mitochondrial VDAC1 to the Plasma Membrane"

_biomolecules, 2023, doi:10.3390/biom13030557_

Round 1

Reviewer 1 Report

In this manuscript, the authors demonstrated that GPR56 knockdown causes inflammation and secretory dysfunction in pancreatic beta-cells. Intriguingly, the secretory dysfunction was mediated by increased VDAC1 expression and its mistargeting to the cellular membrane which causes ATP leakage. These findings are important for the pathophysiology of type 2 diabetes because those islets have been reported to have a reduced expression level of GPR56. The reviewer has several minor comments.

1. Fig 1C: The two asterisks should be placed on the bars showing between-group comparisons.

2. Page 7, line 261: "-cells" lacks "beta".

3. Fig 6G, H: It might be good to test VBIT-4 and VD1-ab on the GPR56-KD EndoC-bH1 cells. 

Author Response

Comments and Suggestions for Authors

In this manuscript, the authors demonstrated that GPR56 knockdown causes inflammation and secretory dysfunction in pancreatic beta-cells. Intriguingly, the secretory dysfunction was mediated by increased VDAC1 expression and its mistargeting to the cellular membrane which causes ATP leakage. These findings are important for the pathophysiology of type 2 diabetes because those islets have been reported to have a reduced expression level of GPR56. The reviewer has several minor comments.

  1. Fig 1C: The two asterisks should be placed on the bars showing between-group comparisons.

Response: This has been corrected.

  1. Page 7, line 261: "-cells" lacks "beta".

Response: This has been corrected and can be found on page 7, line 303 in revised version of manuscript.

  1. Fig 6G, H: It might be good to test VBIT-4 and VD1-ab on the GPR56-KD EndoC-bH1 cells.

Response: Although this is a very interesting point raised by the reviewer, our aim here was to see to what extent the cAMP generation linked to the insulin secretory response of EndoC βH1 cells to glucose and Coll III is affected by GPR56-KD. In view of the limited time at disposal for the revision, the experiments suggested by the reviewer could not be performed.

Reviewer 2 Report

In this study, the authors examined whether GPR56-KD is associated with increased VDAC1 expression and its mistargeting to the cell surface in rodent and human β-cells. The work presented is novel. The authors show GPR56 knockdown is associated with the Coll III-induced suppression of cellular stress-related signal pathways. The data also finds that loss of GPR56 increases VDAC1 expression and subsequent surface mistargeting.

1. GPR56 depletion results in an abolishment of Col1 III actions and an increase in Vdac1 expression. Given that VDAC1 overexpression leads to its mistargeting to the β-cell membrane, the impact of Col1 III on VDAC1 expression and its sub-cellular localization should be examined in Gpr56-KD condition.

2. Fig.1: I recommend to include the information that the activities of various signaling pathways were measured in GPR56-KD cells without stimulation of Col1 III.

3. Fig.4C: The authors show glucose-stimulated insulin secretion is markedly reduced in Gpr56-KD cells compared to scramble controls. However, the biphasic insulin secretory response to 20 mM glucose was similar in islets from WT and Gpr56−/− mice (Ref. 19, PMID: 29855662). Some explanation for this difference from the literature would be useful.  

4. Line 287, Fig.4c is miscited.

Author Response

In this study, the authors examined whether GPR56-KD is associated with increased VDAC1 expression and its mistargeting to the cell surface in rodent and human β-cells. The work presented is novel. The authors show GPR56 knockdown is associated with the Coll III-induced suppression of cellular stress-related signal pathways. The data also finds that loss of GPR56 increases VDAC1 expression and subsequent surface mistargeting.

  1. GPR56 depletion results in an abolishment of Col1 III actions and an increase in Vdac1 expression. Given that VDAC1 overexpression leads to its mistargeting to the β-cell membrane, the impact of Col1 on VDAC1 expression and its sub-cellular localization should be examined in Gpr56-KD condition.

    Response: Since the presence of Coll III did not affect the ATP loss in GPR56-KD β-cells as compared to VDAC1 selective inhibitors, we did not feel it necessary to investigate this (See figure 4 and 5).

  1. Fig.1: I recommend to include the information that the activities of various signaling pathways were measured in GPR56-KD cells without stimulation of Col1 III.

Response: This information has now been included (page 8, line 347-351).

  1. Fig.4C: The authors show glucose-stimulated insulin secretion is markedly reduced in Gpr56-KD cells compared to scramble controls. However, the biphasic insulin secretory response to 20 mM glucose was similar in islets from WT and Gpr56−/−mice (Ref. 19, PMID: 29855662). Some explanation for this difference from the literature would be useful.

Response: We thank the reviewer for pointing out this, indeed this has been discussed on page 11, line 480-483 in the manuscript. 

  1. Line 287, Fig.4c is miscited.

Response: Sorry for that! It has been corrected in the revised version (page 7, line 332).

Reviewer 3 Report

General comments:

As a continuation of the authors’ previous work, with in-vitro setting, the authors provided substantial data that reveals the link between ablation of GPR56 and increased VDAC1 that causes impaired ATP accumulation and insulin secretion in beta cells. I really enjoyed reading the manuscript that is well written with convincing and comprehensive data. 

Specific comments

1.     I am wondering if the mitochondrial function, such as OCR has been assessed throughout this study.  

2.     For the figure 5, is there a specific reason to measure ATP release under non-stimulatory condition (1mM glucose), instead of stimulatory/glucotoxic condition (20mM glucose)? 

3.     VDAC1 protein expression was assessed and quantified by immunofluorescent staining. Is there any western blot data available?

Author Response

As a continuation of the authors’ previous work, with in-vitro setting, the authors provided substantial data that reveals the link between ablation of GPR56 and increased VDAC1 that causes impaired ATP accumulation and insulin secretion in beta cells. I really enjoyed reading the manuscript that is well written with convincing and comprehensive data. 

Specific comments

  1. I am wondering if the mitochondrial function, such as OCR has been assessed throughout this study.

Response: In the present manuscript, we have measured the release and content of ATP in the β-cells. Glucose raises ATP level in β-cells. Since ATP production reflects mitochondrial oxidative phosphorylation, this could be taken as an indirect measure of oxygen consumption (Wiederkehr and Wollheim 2012, reference 8).   

  1. For the figure 5, is there a specific reason to measure ATP release under non-stimulatory condition (1mM glucose), instead of stimulatory/glucotoxic condition (20mM glucose)? 

Response: Since high glucose increases VDAC1 expression and also causes its mistargeting to the cell surface in β-cells we performed the experiments at low glucose to avoid glucose-induced ATP changes.

  1. VDAC1 protein expression was assessed and quantified by immunofluorescent staining. Is there any western blot data available?

Response: For the VDAC1 protein levels upon GPR56-KD in EndoC βH1 cells, we have now performed such Western blots and the results are shown in supplementary figure 2D.